# A Walk through Gumboro Disease

**Maria Pia Franciosini [1],\* and Irit Davidson [2]**

[1] Department of Veterinary Medicine, University of Perugia Study, 06135 Perugia, Italy
[2] Kimron Veterinary Institute, Bet Dagan P.O. Box 12, Israel
\* Correspondence: maria.franciosini@unipg.it

**Abstract:** Infectious bursal disease (IBD), caused by an *Avibirnavirus*, belonging to the family *Birnaviridae*, is an immunosuppressive disease that affects 3–6-week-old chickens, resulting in clinical or subclinical infection. Although clinical disease occurs in chickens, turkeys, ducks, guinea fowl, and ostriches can be also infected. IBD virus (IBDV) causes lymphoid depletion of the bursa, which is responsible for the severe depression of the humoral antibody response, primarily if this occurs within the first 2 weeks of life. IBD remains an issue in chicken meat production due to economic losses caused by the spread of variants or subtypes, resistant to the most common vaccines, responsible for a subclinical disease characterized by reduced growth performance and increased susceptibility to secondary infections. Very virulent strains of classical serotype 1 are also common in several countries and can cause severe disease with up to 90% mortality. This review mainly focuses on the immunosuppressive effect of the IBDV and potential vaccination strategies, capable of overcoming challenges associated with the optimal time for vaccination of offspring, which is dependent on maternal immunity and IBDV variant occurrence.

**Keywords:** IBDV; variants; immunosuppression; subclinical disease; vaccination

## 1. Introduction

Infectious bursal disease (IBD), also known as Gumboro disease, is an immunosuppressive disease that occurs in young chickens between 3 and 6 weeks, resulting in clinical or subclinical infection, both of which are responsible for immunosuppression [1,2]. Gangrenous dermatitis [3], coccidiosis [4], and vaccination failures are frequently associated with IBDV-induced immunosuppression [2].

In 1962, the first case of IBD was reported in Gumboro, Delaware [5]. It spread across the United States and invaded Europe in the 1970s [6]. Control of IBDV infections has been complicated by the recognition of "variant" strains of serotype 1, originating in Delmarva, USA, which caused rapid bursal atrophy without mortality and were capable of evading maternal immunity directed primarily at "classical" strains [7]. These variants or subtypes exhibited different biological properties, compared to classical strains, and could be a consequence of immune pressure due to the extensive application of vaccine plans [8]. Successively very virulent (vv) IBDV strains, responsible for 90% mortality rates, spread to the Netherlands and the United Kingdom in 1988 [9] and then to the rest of world, except Australia, New Zealand, Canada, and the United States until 2008 [10]. Significant differences between vvIBDV strains in Europe and Asia suggest independent IBDV evolution [11]. Jackwood et al. [12] concluded that approximately up to 60% of IBDV isolates worldwide belong to the vvIBDV genotype of the virus. Since then, several studies have addressed the evolution of IBDV around the world, focusing on the emergence of variants [13], recombinant [14,15], and reassortant strains of the virus [16,17].

An Italian IBDV strain (ITA strain), responsible for the subclinical disease associated with a severe immunosuppression status, has been recently detected [18,19]. Whole genome characterization has evidenced that ITA is genetically different from classical IBDV strains.

It has been classified into genogroup 6, together with a few other strains detected in Saudi Arabia and Russia [20].

There is evidence that good management practices, based on the "all in/all out" application, associated with cleaning and disinfection practices of houses, can control the virus infection between production cycles. However, these measures are not definitive without a choice of suitable vaccine programs related to the epidemiological situation of the geographic area. This review aims to provide an oversight of IBDV, mainly focusing on the immunosuppressive effect of the IBDV and the aspects related to the application of a successful vaccine strategy to overcome the obstacles posed by maternal immunity and IBDV variants.

## 2. Etiology

IBDV is a double-stranded non-enveloped RNA virus belonging to the *Birnaviridae* family, *Avibirnavirus* genus [21]. On the basis of virus neutralization tests two serotypes have been recognized: serotypes 1 and 2. Both serotypes can naturally infect chickens, turkeys, ducks, guinea fowls, and ostriches, although only serotype 1 has been reported as pathogenic for chickens [2].

The IBDV genome consists of two segments of double-stranded RNA. The larger fragment, A, encodes viral proteins VP2, VP3, VP4, and VP5, while the smaller fragment, B, encodes VP1, the RNA-dependent RNA polymerase [2]. The conclusive cleavage of VP2, a capsid protein of IBDV containing major immunodominant epitopes and stimulating the production of neutralizing antibodies against IBDV, is of primary importance in the replicative process of the virus [22]. This domain, due to the presence of major hydrophilic peaks, A (212–224 aa) and B (312–324 aa), undergoes possible mutations that can influence virulence, tissue culture adaptation, and antigenic properties of the virus, thus rendering several commercial IBD vaccines ineffective [23,24].

In accordance with their pathogenicity and antigenicity characteristics, IBDVs have been traditionally grouped into four phenotypes: classic, variant, very virulent, and attenuated [25].

With the emergences of novel strains produced by continuous mutations and recombination, defining new IBDV strains with traditional descriptive classification has become increasingly difficult. Consequently, IBDV has been classified into seven genogroups based on the characteristics of the amino acids in the hypervariable region of the capsid protein VP2 of serotype 1 [26]. Recently, Wang et al. [27] proposed a new scheme based on the molecular characteristics of both VP2 and VP1 capsid proteins, encoded by segments A and B, respectively. Following this scheme, IBDV can be categorized into nine genogroups of A and five genogroups of B, and the genogroup A2 can be further divided into four lineages. The classic, variant, very virulent, and attenuated phenotypes correspond to the A1B1, A2B1, A3B2, and A8B1 genotypes.

## 3. Clinical Signs

The acute form of IBDV occurs usually after 2–3 days of incubation in 3–6-week-old chickens, and it is characterized by sudden onset of depression, expressed by head resting on the litter [28], polyuria, ruffled feathers, dehydration, and death. At times, chickens show vent pecking due to the discomfort caused by the increased size of the bursa of Fabricius.

Older birds usually develop subclinical forms, although a recent acute outbreak has been reported in Nigeria in 24-week-old hens vaccinated against IBD [29]. Morbidity varies according to the strain involved and can reach 100% in highly susceptible groups, while mortality rates may peak at 20–30% in outbreaks caused by classical IBDV [2]. Nowadays, the most common form of IBDV infection is subclinical, but the impact on growth performance is still high and mortality can range from 5% to 30%, depending on the affected birds' degree of protection and/or the strains involved. Weight loss and increased food conversion ratio (FCR) have been reported following IBDV-associated immunodeficiency, due to susceptibility to secondary infections [30,31].

## 4. Macroscopic and Microscopic Lesions

Birds suffering from acute-form IBDV are dehydrated and may present petechiae in the pectoral and thigh muscles, likely due to coagulation disorders [32]. The most characteristic lesion is an enlarged bursa with yellowish transudate, which may also cause urate deposits in kidneys, leading to dehydration and/or ureter blockage [33]. Occasionally, hemorrhages are evidenced in proventriculus mucosa and throughout the bursa of Fabricius. Histological examination indicates marked oedema located in the subserosal and interfollicular spaces. As the infection progresses, bursal lymphocyte necrosis also advances towards the cortex. Subsequently, heterophils and reticuloendothelial cells replace necrotic lymphocytes in follicles that at times show the formation of cysts [34]. The proliferation of cortico-medullary epithelium of bursal follicles can create glandular-like intrafollicular structures (Figure 1A,B) [35]. VvIBDV strains are known to produce severe lesions also in non-bursal lymphoid organs, especially in the thymus, spleen, and cecal tonsils, likely due to the action of the virus at these levels [36]. Lymphocytic depletion, both in the cortex and the medulla of the thymus, is caused by apoptosis as well as necrosis, clearly highlighted by electron ultrastructural examination [37] (Figure 2). An immunohistochemical study has revealed virus-antigen positive epithelial reticular cells in the thymus medulla, indicating a possible viral direct action [38], although a TNF (Tumour Necrosis Factor) action could not be excluded [39].

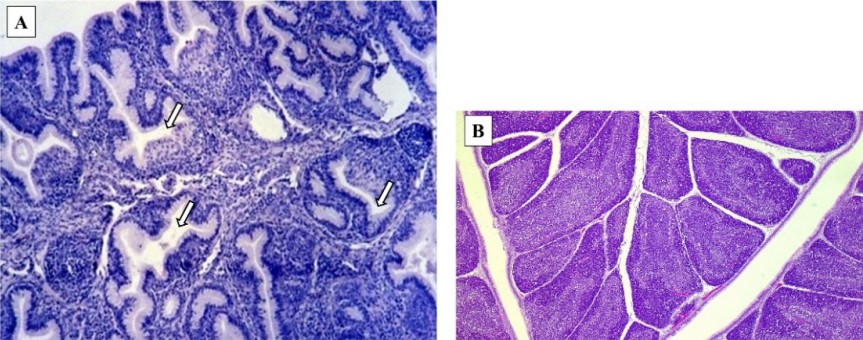

**Figure 1.** Light microscopy. Bursa of Fabricius. (**A**) Glandular-like structures are evident for the proliferation of corticomedullary epithelium of follicles in IBDV affected chickens (arrows) (H&E staining ×200) (**B**). Lymphatic follicles in healthy chickens (H&E staining ×40) [35].

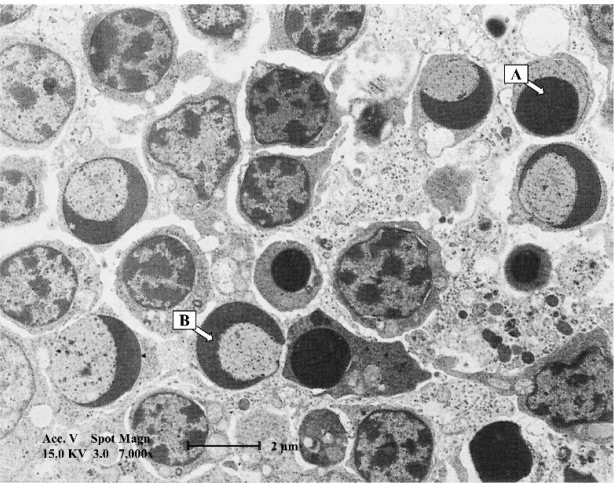

**Figure 2.** Electron microscopy. Thymus. Apoptotic lymphocytes with crescent-like chromatin accumulations beyond the nuclear membrane (**B**). Nuclei transformed in round apoptotic bodies are also evident (**A**) [37].

## 5. Pathogenesis of IBDV

After oral and/or nasal infection, the virus replicates in macrophages and lymphoid cells associated with the gut mucosa [1]. A primary viremia, occurring through portal circulation, leads IBDV in bursal follicles, where an extensive replication is observed in B lymphocytes [40,41] (Figure 3). In particular, the IgM+ B cells serve as the primary targets of IBDV, and surface immunoglobulin M (sIgM) is the cellular receptor, firstly described for IBDV [42]. However, IBDV, as in the case of other non-enveloped viruses without an outer membrane, cannot enter the target cells directly by membrane fusion and different mechanisms as the cellular membrane perforation and conformational alteration have been hypothesized to explain the viral passage across the membrane [43]. In an IBDV infection, a capsid-associated peptide was demonstrated to have permeabilization activity, responsible for producing pores in the endosomal membrane [44,45].

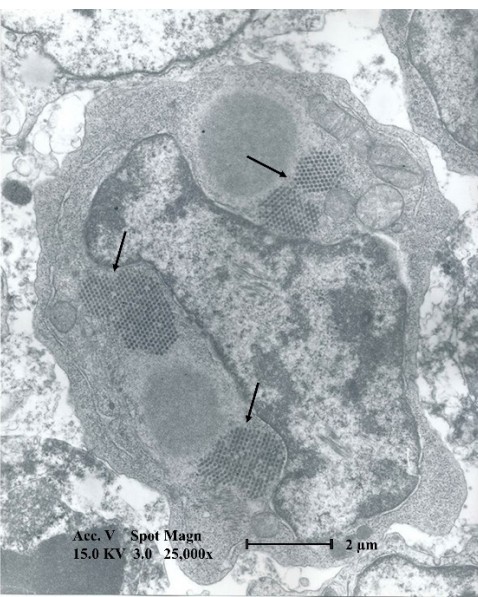

**Figure 3.** Electron microscopy. Bursa of Fabricius. Crystalline-arrayed virus particles in the cytoplasm of a lymphoid cell (arrows) [40].

While classical IBDV serotypes primarily attack immature B lymphocytes, vvIBDV strains affect both immature and mature B lymphocytes [46].

After replication in bursal lymphocytes a second viremia, leading the virus to spread to other organs such as muscle tissue and kidneys, produces clinical signs and death [2,10]. A similar, but more dramatic, trend has been observed in infections caused by vvIBDV strains, producing an increase in mortality, from 50% to 90% compared to classical serotypes, and a more severe state of immunosuppression [47].

Recent studies have shown that the upregulation of proinflammatory cytokines and chemokines, as well as the migration of inflammatory cells, display an unquestionable role in IBDV pathogenesis [48,49]. Chen et al. [50] demonstrated that the vvIBDV, strain NN1172, was able to produce the TLR3-IFN-$\alpha$/$\beta$ pathway, macrophage activation, and the Th1/2 cytokine expression stronger than the B87, a vaccine-attenuated strain.

T lymphocytes, detected in the first stage of infection around the IBDV infected B cells, upregulate gene-expressing cytokines, responsible for macrophage activation and the production of IFN-$\gamma$, TNF, and NO (nitric oxide), exacerbating the bursal lesions [51]. In addition, IFN-$\gamma$ appears to produce apoptosis in infected cells and in healthy B ones surrounding them [52,53].

Eterradossi and Saif [2] observed the dissemination of the virus to other lymphoid organs, such as bone marrow, thymus, spleen, Peyer's patches, cecal tonsils, and Harderian glands in chickens affected by vvIBD. Cecal tonsils and bone marrow could support

a successive replication of IBDV [54]. The presence of maternal antibodies influences pathogenesis in commercial chickens. The virus persisted up to 3 weeks in experimentally infected specific-pathogen-free (SPF) chicks, but a shorter duration was noted in the presence of maternal antibodies [55].

## 6. Immunosuppression

IBDV produces the depletion of B lymphocytes, the leading players in the humoral immune response. In turn, it causes severe immunosuppression, making birds more susceptible to secondary infections and determining a poor vaccine response [56,57]. The immunosuppressive effect of IBDV in response to NDV vaccination has been documented over the years [58,59]. IBDV-infected chickens have also been reported to be more susceptible to other diseases, such as coccidiosis, gangrenous dermatitis, and salmonellosis [60–62]. Although B-lymphocyte re-population in the bursa occurs, the birds display a poor primary antibody response until seven weeks post-infection [33,58]. It was shown that bird age and strain pathogenicity affected bursal recovery [63], and the *in ovo* administration of classical virulent IBDV caused severe depletion and apoptosis of thymocytes [64]. Additionally, the downregulation of CD132+ and CD8+, upregulation of CD132+ and CD25+ T cells in the bursa, and altered secretion and function of cytokines were also observed in the thymus [65]. The recruitment of CD4 and CD8 T lymphocytes also promotes damage in the BF by releasing cytotoxic cytokines, responsible for prolonged immune suppression after IBDV infection [66].

*Acute and Sub-Acute Influences of IBDV Infections in Single- and in Multiple Virus Infections*

IBDV is one of the most known immunosuppressive (IS) viruses of chickens, including also CAV, MDV, REV (Reticuloendotheliosis Virus), and ALV (Avian Leukosis Virus [67–69]. The presence of IS viruses in commercial poultry, especially in an unfavorable management, plays a negative role in growth performances and animal health. IBDV co-infection is seen responsible for the aggravation of the pathogenicity caused by poultry respiratory viruses, such as avian influenza virus, subgroup H9N2, and Newcastle disease virus [70]. Chickens infected with adenovirus and IBDV had more severe pneumonic lesions and tracheitis than birds infected with a single virus [71]. Further, day-old SPF chicks coinfected with IBDV and fowl adenovirus, serotype 4 (FAdV-4), showed increased mortality, enhanced clinical symptoms, and more severe tissue lesions. The expression of interleukin (IL)-6, IL-1β, interferon-γ, and mRNAs in the IBDV and FAdV-4 coinfected chickens was also delayed, and the antibody response levels were significantly lower compared with the FAdV-4 infected chickens, indicating that the IBDV infection could significantly promote the pathogenicity of FAdV-4 and reduce the immune response in chickens [72]. Chicken administrated IBDV vaccine, followed by *S.* Enteritidis infection, could cause severe effect on the bursa of Fabricius, resulting in failure of systemic and mucosal antibody responses to the *S.* Enteritidis and reduce its elimination and clearance [73]. Toro et al. [74] reported that the effects immunosuppressive were even more pronounced in birds infected with more than one IS viruses, as seen in IBDV associated with CAV infection [75] as well as in combined infection with MDV and CAV [76]. Additionally, multiple infections with CAV, IBDV, and adenoviruses, common on chicken farms, are often underestimated, due to the similar subclinical evolution responsible for economic losses and lack of a suitable diagnostic tool application [77]. In this respect, a multiplex RT-PCR assay combined with fluorescence-labeled polystyrene bead microarray (MagPlex-TAG system) has been showed to be a helpful tool to detect multiple infections due to IS such as IBDV, avian reovirus (ARV), CAV, Marek's disease virus (MDV), and reticuloendotheliosis virus (REV) [78].

## 7. Diagnosis

Clinical signs and post-mortem findings in chickens affected by acute form are sufficient for a presumptive diagnosis, although laboratory confirmation is necessary especially

in the subclinical disease, usually characterized by severe atrophy of bursa without the presence of symptoms.

### 7.1. Virus Isolation

Isolation and identification of the virus is the most appropriate diagnostic tool, but it cannot be applied routinely, because it is a time-consuming laboratory procedure. The most sensitive diagnostic method for virus isolation is the inoculation of bursal homogenates from IBDV-infected chickens into the chorioallantoic membrane of 9/10-day-old embryonated SPF chicken eggs. This method is suggested for vvIBDV that is not able to replicate in conventional cell cultures, such as chicken embryo fibroblasts (CEF) or chicken embryo kidney (CEK), unless the virus has previously undergone serial passages in the embryos [79].

### 7.2. Detection of Viral Antigen

The most suitable procedures are agar gel immune diffusion (AGID) and antigen capture enzyme-linked immunosorbent assay (AC-ELISA). Monoclonal antibody use could be helpful in differentiating classic and variants strains. The one-step strip test based on the use on colloidal gold-labelled monoclonal antibodies is recommended for quick diagnosis in field, due to its high sensitivity and specificity [80].

### 7.3. Molecular Methods

Reverse transcriptase polymerase chain reaction (RT-PCR), nucleotide sequence analysis, and multiplex and quantitative real-time RT-PCR (qRT-PCR) are the classic molecular methods used for diagnosis. They are often combined to identify variable regions of the VP2 gene for a more in-depth characterization of IBDV strains [81]. RT- PCR determines the virus load in infected samples and the use of labelled probes further improves this procedure, permitting a reliable differentiation of IBDV strains [82]. The restriction fragment length polymorphism (RFLP) is currently considered an outdated approach since restriction enzymes cannot distinguish accurately the subtypes. Rubinelli and Lin [83] reported that real-time RT-PCR, targeting different regions of the IBDV genome, such as the VP1, VP2, and VP4 genes, in association with melting curve analysis, is able to determine up to a single nucleotide polymorphism and trace the diffusion of vvIBDV strains and of atypical ones. Nowadays, the whole genome-sequencing approach represents a rapid and reliable tool for isolate characterization, which allows for a greater understanding of viral strains circulating in the countries [84,85].

### 7.4. Serological Diagnosis

The agar-gel precipitin (AGP) test or the antigen capture enzyme-linked immunosorbent assay (AC-ELISA) are the most common procedures used in IBDV diagnosis together with the virus neutralization (VN) test. The ELISA is the most widely employed, since it is quick, economical, and adaptive to computer software automation, but VN is considered the gold standard, as it discriminates the antibodies following infections caused by IBDV variants [86]. Serological tests, especially ELISA, are frequently used to determine the effectiveness of vaccine-immune response and the level of maternally derived antibodies (MDAs) [2]. In this regard, recently Gomez et al. [87] showed that the plants (*Nicotiana benthamiana*) can be suitable platform for the production and assembly of subviral IBD particles to be used as a reliable antigen in ELISA test.

## 8. Vaccination Strategies

Vaccination is an essential device for the prevention and control of IBD. Different modified live vaccines (MLVs) have been developed and classified as mild, intermediate, and intermediate plus, according to the degree of virus attenuation. Their effectiveness depends upon their ability to break through maternally derived antibodies (MDAs), which provide protection for chicks in the first 2–3 weeks, although it could interfere with an

early vaccine administration [88]. A half-life of 6.7 days for IBDV-specific MDAs was reported in slow-growing meat chicks and a shorter one (approximately 3 days) in broiler chicks [89]. The MDA level could vary among hatches and among chicks from the same hatches, and it is also influenced by management conditions [90]. The most common method of IBDV prevention consists of the early administration of suitable attenuated live vaccines, which do not interfere with chicks' parental immunity and do not produce significant bursal reactions [91]. Experimental conditions and field trials have shown that humoral immunity following vaccination with a commercially available intermediate IBDV vaccine is strictly associated with the bursal lesions [92,93]. Vaccination success is mostly dependent on the MDA level, which, if it is below the breakthrough level of the vaccine, allows the development of a protective level of IBDV antibodies without considerable bursal lesions. Although agreeing that the administration of intermediate IBDV vaccines was highly effective in SPF chickens, as demonstrated by clinical protection and antibody response, Coletti et al. [92] stated that this protection could be poor in commercial chickens depending on the degree of maternal immunity. On the other hand, vaccine-induced lesions could be not differentiated from those produced by field virus infection [93]. However, despite the limits associated with the use of IBDV-attenuated vaccines, they are commonly administrated in 15/20-day-old chickens via drinking water, when the maternal immunity is expected to be reduced. Recent field studies, performed in India by Ray et al. [94] on the use of different strains of an attenuated live vaccine in commercial broilers, provide a contrast to the common scientific knowledge that the live IBDV vaccine strains can be inactivated or break through maternal immunity, causing permanent damage to the young broiler chicken immune response, although it could be dependent on the vaccine strain. Bursal lesion scores following live IBDV-attenuated vaccine, MB1 (derivative of the IBDV MB strains), were lower in comparison to those reported for the immune complex vaccine (Icx) and the conventionally used live IBDV vaccine (MB group). Additionally, the health status and productive performances were also better in MB1 group.

Inactivated vaccines are usually administered in water-in-oil emulsions, as supporting adjuvant, and in repeated injections to boost up priming vaccination with attenuated live IBDV vaccines. Their use is common for breeder chickens in order to provide immunity to the progeny against early infection with IBDV [95,96]. Thanks to advances in biotechnologies, new vaccines have been developed in recent years to overcome the difficulties related to IBD vaccination and to create successful control strategies.

### 8.1. Subunit Vaccines

The VP2/3/4 polyprotein or VP2 (rVP2) alone has been encoded in different expression systems, such as *Baculovirus* [97,98], *Saccharomyces cerevisiae* [99], *Escherichia coli* [100], *Lactococcus lactis* [101], *Pichia pastoris* [102], or Fowlpox virus [103]. Although experimental trials have demonstrated that these kinds of vaccines can provide good protection, they need to be administered parenterally with adjuvants and recalls must be carried out, which involves additional costs [10,104]. However, recombinant vaccines based on VP2 expressed in *E. coli*, *P. pastoris*, and *Baculovirus* have been commercially licensed and used in some countries [88,102].

### 8.2. DNA Vaccines

Another promising approach is a DNA vaccine based on plasmids expressing the polyprotein gene [105,106] or VP2 gene [107] and able to promote both humoral and cell-mediated immune response with variable efficacy, although resulting in bursal lesions. A priming intervention in ovo or on day-old chicks followed by a booster of inactivated vaccine resulted in satisfactory immune protection in offspring [108]. To improve the effectiveness of this vaccine, the incorporation of cytokines genes, such as IL2, IL6, IL7, and IL18, IF $\gamma$, has been considered [109,110]. The quantity of DNA used in the priming vaccine, the challenge strain of the viruses, bird age, and the way of administration are fundamental factors influencing the vaccine's effectiveness, which is still experimental [104].

*8.3. Immune-Complex Vaccines*

A mention should be reserved for the immune-complex vaccine, consisting of a mixture of IBDV intermediate vaccine associated with antibodies [111]. It is suitable for *in ovo* automated administration by injection through the eggshell on the 18th day of incubation, when eggs are moved to hatching trays. The immune-complex vaccine can provide the so-called "intelligent vaccination" against IBDV, ensuring the development of active immunity in chicks with the continuous release of the virus by dendritic cells and macrophages until the MDAs disappear, avoiding the immunity gap [112,113]. Recently, in order to evaluate possible changes in immunological parameters in SPF chicks vaccinated against infectious bronchitis (IB), following the use of different IBDV vaccines, Lupini et al. [18] showed that the groups administered with an immune-complex vaccine exhibited lower IBDV antibody titers compared to those given a vaccine consisting of a dual recombinant herpes virus of turkey (rHVT) expressing both VP2 protein of IBDV, and F protein of Newcastle disease virus.

*8.4. Live Viral Vector Vaccines*

Vector vaccines are genetically engineered vaccines in which a gene from one organism genome (donor) is inserted into another organism genome (vector) to produce an active immune response to both agents [114]. Concerning IBDV, several viruses have been used as vectors of the capsid protein VP2: *Baculovirus* [98], *Avian adenoviruses* [115], and *Herpesvirus* of turkey (HVT) [116]. In this respect, HVT has been successfully used, even in the presence of MDAs [117,118]. Prandini et al. [118], in particular, studied the effect of the HVT vector vaccine (vHVT-IBD) on circulating B cells and the ability to induce protection against vvIBDV challenge application in commercial pullets, comparing an intermediate and intermediate plus IBD vaccine based on the strains D78 and 228E, respectively. IBD live-vaccinated groups showed significantly percentages of circulating B cells lower than vHVT-IBD and non-IBD-vaccinated groups. Moreover, ELISA antibody levels against NDV, IBV, and EDS were considerably higher in the vHVT-IBD and non-vaccinated control groups compared to those observed in IBD-intermediate plus-vaccinated and intermediate-IBD vaccine groups. After the vvIBDV challenge, the virus was detected by qRT-PCR in the bursa tissue of all vvIBDV-challenged birds, but the most predominant virus in the bursa of Fabricius of IBD live-vaccinated pullets was the vaccine strain. In contrast, the vvIBDV challenge strain was prevalent in the SPF and vHVT-IBD-vaccinated and challenged birds, suggesting that the IBD live vaccine may control vvIBDV replication by direct competition toward the same target cell receptors or by a more effective activation of innate immunity, as speculated by Ramon et al. [119].

Over the years, a number of VP2-based HVT-IBD vector vaccines, applied in *ovo* or via the subcutaneous/intramuscular route in day-old chicks, have been developed and licensed in various countries, and data on field efficacy have been reported [120,121].

Vaccinated chickens produce anti-VP2 antibodies that appear to be effective against challenges by classic, variant, and vvIBDV strains [122,123]. Nowadays, HVT-IBDV vaccines seem to be the most suitable alternative to conventional IBD live vaccines since they do not interfere with MDAs and have an improved safety profile compared with live vaccine.

## 9. Conclusions

Years after its appearance, IBDV is still a problem causing economic losses in the poultry industry worldwide. The extensive application of vaccination programs has created a selective pressure responsible for the onset of an unexpectedly wide variety of variants, as recently reported in Central Europe [124,125], some of which with established phenotypical associations, producing a further challenge in the choice of suitable vaccines. Severe forms of the disease, caused by vvIBDV strains, have been also reported worldwide, although the subclinical IBD is more common. Moreover, some epidemiological aspects related to the spread of the virus, such as the possible roles of vectors, e.g., dogs and wild birds [126,127],

are not fully clarified. Although numerous vaccine solutions have been developed over the years, the goal of an optimal large-scale vaccine strategy has not been achieved yet. It is quite obvious that the choice and application of vaccines should be dependent on the diagnostic screening to assess the predominant variants in a defined geographical area and the possible presence of other immunosuppressive viruses, as well as a strict application of biosecurity standards to control the virus spread.

**Author Contributions:** Conceptualization, M.P.F. and I.D.; writing—original draft preparation, M.P.F.; writing review and editing, M.P.F.; visualization, M.P.F. and I.D.; supervision, M.P.F. and I.D. All authors have read and agreed to the published version of the manuscript.

**Funding:** This research received no external funding.

**Data Availability Statement:** Not applicable.

**Acknowledgments:** We thank G. Guelfi, L. Musa, G. Asdrubali, and L. Leonardi for precious collaboration and technical assistance.

**Conflicts of Interest:** The authors declare no conflict of interest.

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
