# Peer review of "A Walk through Gumboro Disease"

_poultry, doi:10.3390/poultry1040020_

Round 1
Reviewer 1 Report
This papers reviews the current knowledge on the main aspects of Infectious Bursal Disease (IBD) and potential vaccination strategies.
The authors give a good review on the current knowledge of the disease and Research and Development currently ongoing on future vaccines / control strategies. However, there is little new information that has not been provided by the Infectious Bursal Disease chapter from Diseases of Poultry 14th edition published on 2019. There is only 12 new citations in the review paper from 2019 onwards that are usually cited together with citations from before 2019. It is my opinion that there is a lack of new or novel points of view on the disease that is offered by the authors in this review that has not been already addressed by Diseases of Poultry 2019.
Perhaps the authors would like to add the current situation in Italy or Israel together with unpublished data. This would offer new and interesting data to this paper. As it is stands now, it offers little new information. Some comments:
Lines 60-61: Incomplete idea. Rewrite sentence "At times..."
Lines 102-104: Add citation
Lines 105-106: Rephrase sentence.
Lines 138-140: It is recommended to either change "poultry" for "chickens", or to add immunosuppressive diseases of Turkeys (like HEV), as well.
Line 144: Rephrase for clarity : "dually virus-infected..."
Lines 160-162: Rephrase for clarity.
Lines 184-188: Please comment that RFLP is rarely used anymore in routinely IBD diagnostics.
Lines 188-191: Please reconsider. Nowadays, it would be preferable to perform sequencing rather than conducting qRT-PCR on three different targets with melting-curve analysis with data from 2007.
Lines 263-276: No comment on viral shedding upon challenge is offered. No comment on field studies are offered (example Ray et al 2021).
Line 278 - Rephrase for clarity
Lines 286-289 - Rephrase for clarity
Author Response
Firstly, we would like to thank you for the time spent on careful revision of our manuscript and your appreciable comments, contributing to improve our work. We hope the revised revision addresses your concerns. Please consider that some lines in reporting the rephrased sentences may not exactly match. Please find attached a point-by point response to your remarks.

Reviewer 2 Report
The authors present a relatively short but well rounded review of the topic that should be of interest to workers in the fields of veterinary infectious diseases, poultry sciences, and virology in general. It is written at a level appropriate so that it is comprehensible for readers from those fields (as well as related disciplines). I have only a couple of suggestions for improvement.
There are still several grammatical errors. These should be corrected.
The histological lesion shown in Fig. 1A: it will be helpful to mark the glandular structures (i.e. one of them) with an arrow and to show a micrograph from a healthy control for comparison.
Add 2-3 additional sentences providing some more detail about the viral life cycle. Is it known how the virus enters host cells ? (is the receptor known?) Compared to enveloped viruses, is there anything of importance regarding vaccine development in these non-enveloped viruses?
Author Response
We would like to thank you for helpful comment. We hope the changes made can satisfy your requirements. Please find attached the point-by-point answers to your remarks

Reviewer 3 Report
The review of Franciosini and Davidson broadly covers the problems of infectious bursal disease: classification of viruses, clinical presentation, diagnosis, strategy and problems of vaccination, and characteristics of different vaccines. The problem of the influence of maternally derived antibodies on the success of vaccination has been well described.
Author Response
We would like to thank you for the time spent in reviewing our manuscript and for positive feed back.

Reviewer 4 Report
Economic losses are substantial due to Gumboro Disease (Infectious bursal illness), an immunosuppressive disease that affects young chickens (aged 3 to 6 weeks). The author conducts a literature analysis on the most important components of the illness, zeroing in on prospective vaccination tactics that might help overcome the challenges of selecting the most ideal period for immunization of chicks owing to the existence of maternal immunity and IBDV variations. I feel the current research work style is suitable for publication in MDPI's Poultry journal. So, I suggest accepting the article with minor adjustments. But first, here are some general and specific comments to the authors
General comments:
The chicken meat sector continues to suffer losses because IBD (infectious bursal disease) is still an issue despite attempts to combat it. The author focused mostly on immunization tactics, some of which are novel to this context; the study work style is appropriate for MDPI's Poultry publication.
Specific comments:
1. Introduction: The introduction should be rewritten and should be highlighted in a summary in the last Para of the introduction.
2. In lines 235 and 245, please fix the typos.
3. The author could make a table based on various vaccines from a different country
3. The English language and style must be moderately edited
Author Response
We would like to thank you for your thoughtful comments and efforts towards improving our manuscript. We hope we have addressed your concerns. Please find attached file including answers to your comments.
